# Phenolic Compounds Cannabidiol, Curcumin and Quercetin Cause Mitochondrial Dysfunction and Suppress Acute Lymphoblastic Leukemia Cells

**DOI:** 10.3390/ijms22010204

**Published:** 2020-12-28

**Authors:** Miguel Olivas-Aguirre, Liliana Torres-López, Igor Pottosin, Oxana Dobrovinskaya

**Affiliations:** Laboratory of Immunobiology and Ionic Transport Regulation, Centro Universitario de Investigaciones Biomédicas, Universidad de Colima, Av. 25 de Julio 965, Villa de San Sebastián, 28045 Colima, Mexico; miguel.a.olivas@gmail.com (M.O.-A.); lilianatorres4667@gmail.com (L.T.-L.)

**Keywords:** acute lymphoblastic leukemia, cannabidiol, curcumin, quercetin, mitochondria, cytotoxicity

## Abstract

Anticancer activity of different phenols is documented, but underlying mechanisms remain elusive. Recently, we have shown that cannabidiol kills the cells of acute lymphoblastic leukemia (ALL) by a direct interaction with mitochondria, with their consequent dysfunction. In the present study, cytotoxic effects of several phenolic compounds against human the T-ALL cell line Jurkat were tested by means of resazurin-based metabolic assay. To unravel underlying mechanisms, mitochondrial membrane potential (∆Ψ_m_) and [Ca^2+^]_m_ measurements were undertaken, and reactive oxygen species generation and cell death were evaluated by flow cytometry. Three out of eight tested phenolics, cannabidiol, curcumin and quercetin, which displayed a significant cytotoxic effect, also dissipated the ∆Ψ_m_ and induced a significant [Ca^2+^]_m_ increase, whereas inefficient phenols did not. Dissipation of the ∆Ψ_m_ by cannabidiol was prevented by cyclosporine A and reverted by Ru360, inhibitors of the permeation transition pore and mitochondrial Ca^2+^ uniporter, respectively. Ru360 prevented the phenol-induced [Ca^2+^]_m_ rise, but neither cyclosporine A nor Ru360 affected the curcumin- and quercetin-induced ∆Ψ_m_ depolarization. Ru360 impeded the curcumin- and cannabidiol-induced cell death. Thus, all three phenols exert their antileukemic activity via mitochondrial Ca^2+^ overload, whereas curcumin and quercetin suppress the metabolism of leukemic cells by direct mitochondrial uncoupling.

## 1. Introduction

Cancer represents a main cause of morbidity and mortality worldwide. Acute lymphoblastic leukemia (ALL) is an aggressive hematologic disorder that occurs mainly in children and adolescents. T-lineage ALL (T-ALL) represents a clinical challenge due its multiple mechanisms of chemotherapy resistance and cell death evasion, responsible for patient chemotherapy failure, relapse and death [1,2]. Therefore, the search for novel antileukemic compounds with a high anticancer effectiveness and low side effects continues.

Phenolic compounds are a group of phytochemicals, containing one or several aromatic rings, which are commonly obtained from plants, vegetables and common beverages such as beer, red wine or coffee. They received attention primarily due to their antioxidant activity. Yet, phenolic compounds possess a broader spectrum of action, including cytotoxic effects in different cancer types [3,4,5]. Previous studies have suggested that high consumption of phenols may reduce cancer risks. Related mechanisms include, but are not restricted to, modifications in the antioxidant system, receptor-mediated cell signaling, cell cycle modifications and cell death induction. Importantly, the anticancer effect of phenols seems to be highly dependent on phenol species and cancer type [6].

Antileukemic activity was demonstrated for several phenolic compounds, albeit precise step-by-step mechanisms remain elusive. Reported effects include a decrease in cancer cells population in vitro and in vivo, deregulation of the Bcl-2 protein ratio, caspase activation, reactive oxygen species (ROS) production, cytochrome c release and apoptosis induction (Appendix A).

Recently, targeting to mitochondria, which provokes their dysfunction, was demonstrated for several small molecules containing phenolic groups, such as ellagic acid, curcumin, aspirin and cannabidiol (CBD) [7,8,9,10]. Several phenols with a pKa within the physiological range possess protonophore activity, thus efficiently dissipating the electrochemical gradient for H^+^ across the inner mitochondrial membrane and, in this way, suppressing different cancer cell types [11]. Notably, leukemic mitochondria have been proposed as an attractive target for anticancer therapy. Mitochondria acquire different alterations during malignant reprogramming that make them susceptible to small molecules with anticancer activity, known as mitocans, which include phenolic compounds [12,13].

We have recently demonstrated that CBD, a main phytocannabinoid derived from *Cannabis* spp., kills leukemic cells by directly targeting mitochondria and causing ROS generation, mitochondrial calcium ([Ca^2+^]_m_) overload, stable mitochondrial permeability transition pore (mPTP) formation and cytochrome c release, which eventually promotes apoptosis and mPTP-driven necrosis [10]. Western blot data and experiments with isolated voltage-dependent anion channel (VDAC) protein, incorporated into a planar lipid bilayer, demonstrated that CBD directly interacts with and switches VDAC to a closed conformational substate [14]. A similar mechanism was also reported for curcumin and aspirin and is considered to be the cause of the tumor cell death [8,9]. VDAC is unique porin, functionally present and abundant in the outer mitochondrial membrane. It acts as the mitochondrial gatekeeper, mediating ionic and metabolic fluxes between the cytosol and the mitochondrial intermembrane space [15,16,17]. The selectivity of this exchange critically depends on the conformational state of VDAC. The aforementioned closed conformational substate is impermeable for large metabolites like adenine nucleotides, but highly permeable to Ca^2+^ [15]. The combination of these two factors can eventually lead to mitochondrial Ca^2+^ overload [10].

The purpose of the present work was to test the cytotoxic effects of several phenolic compounds with a documented antileukemic activity, using the human Jurkat cell line as a model for acute lymphoblastic leukemia of T type (T-ALL). Specifically, we wish to unravel whether the anti-T-ALL activity of the phenolics is correlated with their effects on ROS generation and alterations of the mitochondrial parameters, such as electrical potential difference and Ca^2+^ homoeostasis, and whether the prevention of the latter can revert the cytotoxicity.

## 2. Results

### 2.1. Comparison of Antileukemic Properties of Different Phenolic Compounds in T-ALL Model

Eight phenol-containing compounds that were reported earlier to possess cytotoxic properties against different types of leukemia (Appendix A) were selected to compare their antileukemic potential against T-ALL-derived Jurkat cells. A metabolic activity assay showed that CBD and curcumin were the most cytotoxic, with an IC50 of 12.1 and 36.5 µM, respectively (Figure 1). Chlorogenic and gallic acid, as well as quercetin, exhibited a mild cytotoxicity, whereas aspirin, methyl gallate and protocatechuic acid at concentrations up to 2.5 mM lacked any substantial effect at 24 h.

### 2.2. Curcumin is Rapidly Taken Up by Mitochondria

Some phenolic compounds cause cell death by targeting mitochondria [4,10,17]. We took the advantage of the intrinsic fluorescence of curcumin (Figure 2a) to monitor its uptake and subsequent subcellular localization [18]. Jurkat cells incorporated curcumin rapidly after 10 min of treatment (Figure 2b). Confocal microscopy assays revealed that curcumin was selectively localized in discrete puncta in cytosol (Figure 2c). To determine whether these regions correspond to mitochondria, cells were stained with the mitochondrial-selective dye Mitotracker Red (MtRed) prior to the curcumin treatment. High co-localization of MtRed and curcumin fluorescence (Figure 2d–e) indicates that mitochondria are primary targets for curcumin. Thus, the effect of phenolic compounds on the transmembrane electrical potential difference, whose magnitude reflects the mitochondrial energized status, was further addressed.

### 2.3. Cytotoxic Phenols Promote Mitochondrial [Ca^2+^] Overload and ∆Ψm Loss

We previously found out that CBD evoked mitochondrial Ca^2+^ ([Ca^2+^]_m_) overload, resulting in a stable mitochondrial mPTP formation, ∆Ψ_m_ loss and cell death [10]. These phenomena have also been observed in cells of other types of cancers that were exposed to various phenols [9,19,20,21]. In addition to CBD, curcumin and quercetin, which also demonstrated the capacity to compromise Jurkat cell viability (Figure 1), promoted an immediate [Ca^2+^]_m_ overload (Figure 3a–b). Contrary to these three phenolic compounds, aspirin or chlorogenic acid, which did not exhibit a significant cytotoxicity, provoked an insignificant [Ca^2+^]_m_ rise (Figure 3a–b).

To compare the effect of different phenolic compounds on ∆Ψ_m_, tetramethylrhodamine ethyl ester perchlorate (TMRE) fluorescence was evaluated in TMRE-stained cells (Figure 3c). In line with their cytotoxic effects, CBD, curcumin and quercetin produced an immediate ∆Ψ_m_ collapse (cf. positive control with a classical uncoupler, CCCP). The depolarized state persisted after 4 and 8 h (Figure 3e,g,k), whereas chlorogenic acid and methyl gallate only slightly diminished ∆Ψ_m_ at 8 h (Figure 3f,i), and protocatechuic acid had no significant effect at 4–8 h (Figure 3j). Notably, aspirin and gallic acid favored the TMRE retention (Figure 3d,h), which implies a hyperpolarization instead of a depolarization.

### 2.4. Mitochondrial Uncoupling and Ca^2+^ Overload Mediate the Cytotoxic Effect of Bioactive Phenols

∆Ψ_m_ loss can be attributed to the mPTP formation or the protonophore activity of phenols. To determine the dependence of mPTP-mediated ∆Ψ_m_ loss, Jurkat cells were stained with TMRE and preincubated with the mPTP inhibitor CsA (10 µM, 20 min preincubation). Under this condition, mPTP inhibition suppressed the CBD effects on ∆Ψ_m_. However, the depolarization induced by curcumin and quercetin was unaffected in the presence of CsA (Figure 4a). Inhibition of [Ca^2+^]_m_ overload by mitochondrial Ca^2+^ uniporter, MCU, blocker Ru360 (1 µM, 20 min preincubation) reverted the ∆Ψ_m_ loss induced by CBD but not that by curcumin or quercetin (Figure 4b,c). Thus, curcumin and quercetin may act as direct uncouplers, whereas [Ca^2+^]_m_ overload was necessary for the mPTP formation and ∆Ψ_m_ dissipation in CBD-treated cells. To evaluate the effect of [Ca^2+^]_m_ overload inhibition on cell death induced by phenols, Jurkat cells were treated with CBD (25 µM), curcumin (200 µM) or quercetin (1.5 mM) for 1 h and stained with DAPI as a cell death marker. Cytotoxic effects were rapidly observed for CBD- and curcumin-treated cells (Figure 4d) and were to a great extent prevented by Ru360. Metabolic activity of Jurkat cells was decreased by all three phenols, but only in the case of CBD did Ru360 display a significant protection (Figure 4e), in agreement with previous results [10].

### 2.5. Phenols Differentially Regulate ROS Production

Alteration of the ∆Ψm, by either depolarization, associated with the mPTP opening, or hyperpolarization, due to a hyperactive oxidative phosphorylation, both cause an increase in ROS generation by mitochondria, which can eventually lead to cell death [22,23,24]. On the other hand, phenols per se can act as ROS scavengers. To assess the functional impact of phenolic compounds on the net ROS production, cells were treated with phenols for 1 and 2 h and ROS production was evaluated by flow cytometry using 2′, 7′-Dichlorodihydrofluorescein diacetate, DCFH-DA. Under our experimental conditions, aspirin, CBD and curcumin triggered an elevation in the ROS production as compared to untreated cells (Figure 5a–d). This ROS elevation was transient and undetectable at 2 h post-administration. Meanwhile, gallic acid, methyl gallate, protocatechuic acid and quercetin tend to decrease basal ROS levels, thereby demonstrating antioxidant properties (Figure 5e–h). Apparently, pro- or antioxidant activity of phenolic compounds per se may not explain their cytotoxicity.

## 3. Discussion

Our results supported the view that the antileukemic activity of phenolic compounds is due to their interaction with mitochondria in two ways: a direct uncoupling effect (curcumin, quercetin) (Figure 1a,b,e) and mitochondrial Ca^2+^ overload (Figure 3a,b and Figure 4d). Little correlation was found with their pro- or antioxidant activity (Figure 5), albeit the contribution of the former one to the cytotoxicity of CBD and curcumin is not excluded, whereas the antioxidant activity of quercetin may underlie the reversal of cytotoxicity observed at high concentrations of this compound (Figure 1h).

The uncoupling effect of curcumin and quercetin are in line with their predicted protonophore activity [11]. The data in Figure 4e clearly indicate that these compounds suppressed cell metabolic activity and this effect was hardly reverted by Ru360, which inhibits Ca^2+^ uptake mediated by MCU. However, Ru360 prevented CBD- and curcumin- induced cell death (Figure 4d). Thus, mitochondrial Ca^2+^ overload is essential for the execution of the cell death scenario. A large increase in mitochondrial Ca^2+^, like that induced by CBD, resulted in a stable mPTP formation, collapse of ∆Ψ_m_, release of pro-apoptotic factors and, eventually, cell death [10,14]. On the other hand, a moderate increase in [Ca^2+^]_m_ should stimulate oxidative phosphorylation [25], which may be the cause of the ∆Ψ_m_ hyperpolarization (Figure 3d) and an increase in the ROS production (Figure 5a), induced by aspirin.

We cannot discard the possibility that bioactive phenols directly activate the MCU, thus promoting the mitochondrial Ca^2+^ overload. On the other hand, those phenolic compounds, which cause a moderate (aspirin) or large (CBD, curcumin) increase in mitochondrial Ca^2+^ were shown to directly interact with VDAC1, favoring its closed conformation [8,9,10,14], which is preferentially permeable to Ca^2+^ but excludes adenine nucleotides, like ATP [13,26]. We have tested possible protein–ligand interactions between hVDAC1 and different phenols in silico by means of Molegro Virtual Docker. All phenols appear to interact with hVDAC1 within a conserved pocket or cavity, which includes the N-terminal α helix and β9-13 strands (Appendix A). Of note, the efficient phenols CBD and curcumin yielded a large number of potentially interacting residues, hence a higher overall binding energy, whereas CBD, curcumin and quercetin had the strongest preference for the two specific residues, His 184 (β12 strand at the pore wall) and Thr 9 (α helix in the N-terminus) (Appendix A). This may be not just coincidental. Such an interaction may compete with the ligation between certain residues in the N-terminus and a different region in the pore wall, which is thought to fix the VDAC pore fully open [27,28] thus destabilizing the open state vs. the closed one. Alternatively, or additionally, cytotoxic effects of phenols may be due to the detachment of hexokinase from VDAC, which is also induced by hVDAC1 closure or due to the hVDAC1 oligomerization [13,29].

In a conclusion, natural phenols that exhibited the most promising antileukemic effect were CBD and curcumin. These compounds induce the death of T-ALL cells principally via mitochondrial Ca^2+^ overload. Curcumin also exerts an uncoupling effect. Due to its general physicochemical mechanism, the latter will also affect mitochondria in healthy cells. Thus, it remains to be elucidated whether such a dual action would be beneficial or detrimental for T-ALL treatments. As with any small molecules, the aforementioned compounds have multiple cellular targets. For instance, curcumin can positively modulate the extrinsic and intrinsic apoptotic pathways in several types of cancers and enhance the effects of anticancer drugs [30]. Curcumin also inhibits several K^+^ channels, including those expressed in lymphocytes and leukemic cells, Kv1.3, Kv11.1 and KCa3.1, and the main lymphocyte Ca^2+^ influx channel CRAC modulates chloride channels, ATP-binding cassette (ABC) and glucose transporters [31]. These effects may be crucial for other cancer types. The efficient inhibition of ABC transporters, including multidrug resistance pumps, by curcumin and by flavonoids (quercetin) [32,33], may have an additional therapeutic effect, by the promotion of the intracellular accumulation of other anticancer drugs. Feasibility and limitations of the clinical potential for each compound must be addressed. For example, the legal status of CBD usage as well the high immunosuppressive effects of curcumin against non-oncological lymphoid cells [34] need to be considered. Another problem is the bioavailability of CBD and curcumin. Published data on their pharmacokinetics revealed a maximal concentration in serum of a few micromoles [35,36]. We see two possible ways to overcome this problem. The first is to develop the chemical derivatives of CBD and curcumin, which will act at lower concentrations. For instance, drugs can be tagged in such a way that make them mitochondria targeted [37]. Another non-exclusive approach is to use the advantage of the drugs’ synergism, which can substantially reduce their efficient antileukemic concentration.

## 4. Materials and Methods

(For technical details please consult Appendix A).

### 4.1. Reagents

Reagents used in this study were purchased from Cayman Chemicals (Ann Arbor, Michigan, USA): CBD (90081), chlorogenic acid (70930), gallic acid (11846), methyl gallate (19951), protocatechuic acid (14916), quercetin (10005169); Sigma-Aldrich (San Luis, Missouri, USA): aspirin (A2093), curcumin (C7727), dichlorodihydrofluorescein diacetate (DCFHDA, D6883), phorbol 12 myristate 13 acetate (PMA, P8139); Thermo Fisher Scientific (Waltham, Massachusetts, USA): 4′,6-diamidino-2-phenylindole, dihydrochloride (DAPI, D1306), tetramethylrhodamine, ethyl ester, perchlorate (TMRE, T669), Rhod-2 AM, cell permeant (R1244); Merck (Darmstadt, Germany): Ru360 (557440). Phenolic compounds were prepared as stock solutions and maintained at −20 °C until use: aspirin (134 mM in DMSO), CBD (32 mM in methanol), chlorogenic acid (100 mM in DMSO), curcumin (70 mM in DMSO), gallic acid (150 mM in ethanol), methyl gallate (150 mM in DMSO), protocatechuic acid (250 mM in ethanol), quercetin (150 mM in DMSO). The range of working concentrations used for each drug was in accordance with data available in the literature (Appendix A). The highest final solvent concentrations (in *v*/*v*: 1% DMSO, 0.6% methanol, 0.8% ethanol) did not affect the viability of cells. The effect of phenolic compounds on pH was verified and no significant changes were observed up to the highest concentration tested.

### 4.2. Cells and Culture Conditions

Jurkat cell line (clone E6-1, TIB-152) was obtained from the American Type Culture Collection (ATCC, Manassas, VA, USA). Cells between the 3rd and 20th passage counted from the day of receipt were used. Cells were cultured in suspension in a humidified incubator in 5% CO_2_ atmosphere at 37 °C. Culture medium was Advanced RPMI 1640 supplemented with 5% (*v*/*v*) of heat-inactivated fetal bovine serum (FBS), 2 mM GlutaMAX, 10 mM HEPES and 100 U/mL penicillin–100 μg/mL streptomycin (all from Gibco, Thermo Fisher Scientific, Fairport, NY, USA).

### 4.3. Viability Assay

A resazurin-based metabolic assay in vitro toxicology assay kit Tox8 (Sigma-Aldrich, St. Lois, USA) was used. In this method, non-fluorescent resazurin is reduced to strongly fluorescent resorufin by viable cells. Cells were collected, centrifuged and resuspended in fresh medium (10^6^ cells/mL). The cell suspension (100 µL/well) was placed into a 96-well plate and phenolic compounds were added (in 80 µL/well). After 20 h of incubation, 20 µL of resazurin reagent was added to each well for a total volume of 200 µL. Cells were further incubated for 4 h and the viability was evaluated by measurement of resorufin fluorescence using a GloMax (Promega, Madison, WI, USA) plate reader (Ex: 525 nm, Em: 580–640 nm). Results from independent experiments were averaged and normalized to controls.

### 4.4. Curcumin Uptake Monitoring

Curcumin is a natural yellow-orange dye, derived from *Curcuma longa*, with previously characterized fluorescent properties [18]. Curcumin fluorescence was measured spectrofluorometrically in a quartz cuvette containing 25 µM of curcumin in PBS. Samples were excited at 405 nm and the emission wavelength was scanned (405–600 nm) to determine the lambda maximum. The fluorescence from 3 independent samples was averaged and plotted. To monitor curcumin uptake, Jurkat cells (1 × 10^6^/mL) were exposed to curcumin (1–100 µM) for 10 min. Upon incubation, cells were washed twice with PBS to remove excess dye, transferred to a 96-well plate and fluorescence was estimated by exciting the samples at 405 nm and collecting the emission fluorescence from 500–550 nm in a GloMax Discover (Promega, Madison, WI, USA) plate reader. To determine subcellular curcumin localization, Jurkat cells were pre-stained with 200 nM of Mitotracker Red (MtRed) for 25 min, washed, suspended in PBS and placed in a custom-made coverslip bottom chambers and evaluated by confocal microscopy (LSM 700, ZEISS microscope). Time-series mode with frame acquisition each minute was used. A single cell was selected and focused and basal fluorescence was acquired prior to curcumin addition (5 frames). Next, curcumin (25 µM) was added and images were acquired for the next 10 min. Raw images were further analyzed in ZEN (ZEISS, Munich, Germany) software to split channels and estimate the fluorescence distribution.

### 4.5. Mitochondrial Calcium [Ca^2+^]m Measurements

Jurkat cells were harvested, washed to remove the culture medium and re-suspended in Hanks’ balanced salt solution (HBSS; NaCl 143 mM, KCl 6 mM, MgSO_4_ 5 mM, CaCl_2_ 1.5 mM, HEPES 20 mM, BSA 0.1%, glucose 5 mM, pH 7.4, ≈300 mOsm) for further incubation with the Ca^2+^-sensitive probe Rhod-2AM (2 µM; 30 min; Waltham, MA, USA). Rhod-2 is a cationic dye that preferentially localizes in undamaged mitochondria. After the incubation period, cells were washed to remove extracellular excessive dye, re-suspended in HBSS and assayed 15 min later. Stained cells (1 × 10^6^/mL) were placed in a quartz cuvette. Samples were excited at 552 nm and the Rhod-2 fluorescence was measured at 581 nm using a spectrofluorometer (Hitachi High-Technologies F7000, Hitachinaka, Japan). Changes in [Ca^2+^]_m_ were evaluated as Rhod-2 fluorescence related to initial fluorescence (F/F0).

### 4.6. Evaluation of Mitochondrial Membrane Potential (∆Ψm)

Jurkat cells were collected, culture medium was removed, and cells were resuspended in HBSS. Tetramethylrhodamine ethyl ester perchlorate (TMRE, Ex/Em max = 555/582 nm; Thermo Fisher, Waltham, Masschusetts, USA, T669) is a cationic dye that is sequestered and retained by energized mitochondria. Jurkat cells stained with TMRE (200 nM) were exposed to phenolic compounds for 4 or 8 h and changes in TMRE retention was estimated by measurement of fluorescence intensity in a GloMax Discover (Promega, Madison, WI, USA) plate reader. Samples were excited at 549 nm and emission was collected at 575 nm. HBSS or phenols autofluorescence was subtracted and data from independent experiments were averaged and expressed as % to control group.

### 4.7. Cell Death Analysis by DAPI Retention

Jurkat cells (1 × 10^6^/mL) were preincubated with Ru360 (1 µM) for 20 min in RPMI medium in a 48-well plate. After incubation, cells were treated with the selected phenols for 1 h. Right after, cells were collected, washed and resuspended in PBS, then DAPI was added (1 µM) and incubated for 30 min. Then, cells were washed with PBS to remove excessive dye and DAPI retention was evaluated in a 96-well plate using a GloMax Discover plate reader (Promega, Madison WI, USA) by exciting the samples at 365 nm, while emission was collected at 415–445 nm. Curcumin and quercetin autofluorescence was measured and subtracted to treated cells. Data from independent experiments were averaged and normalized to control group.

### 4.8. Measurement of ROS by Flow Cytometry

To evaluate the increase or decrease in ROS production caused by the treatment of Jurkat cells with phenolic compounds, the non-fluorescent dye 2′, 7′-Dichlorodihydrofluorescein diacetate (DCFH-DA; D6883 Sigma, St. Lois, MO, USA) was used. DCFH-DA is changed to DCF, which is highly fluorescent, when oxidized by intracellular ROS. Jurkat cells were seeded in RPMI cultured medium and treated at indicated concentrations of phenolic compounds by 1–2 h, harvested, washed with PBS and stained with DCFH-DA (100 µL DCFH-DA 5 µM/5 x10^5^ cells) for 30 min at 37 °C. DCF fluorescence (Ex/Em: 495 nm/529 nm) was measured by flow cytometry (FACS Canto II, BD Biosciences, Franklin Lakes, NJ, USA) to evaluate changes in intracellular ROS. In these experiments, a 488 laser and a combination of a 502LP mirror and 530/30 filter were used. Debris and doublets were gated out. Ten thousand events (single-cell gate) were collected for each sample. Data analysis was performed with FlowJo 10.2 Software (BD, Ashland, OR, USA) and median intensity fluorescence (MFI) normalized to control was graphed. Cell and phenol autofluorescence was subtracted from DCF-treated and labeled cells.

### 4.9. Protein–Ligand Interaction Prediction

Molegro Virtual Docker 6.0 software (Odder, Denmark) was used to explore the possible interaction between the phenolic compounds and the human VDAC1 channel. The Pubchem database (NIH) was used to obtain the chemical structure of every phenol studied: aspirin (2244), CBD (644019), chlorogenic acid (1794427), curcumin (969516), gallic acid (370), methyl gallate (7428), protocatechuic acid (72), quercetin (5280343). The hVDAC1 channel structure was downloaded from the protein data bank (PDB, 2JK4). Cavities or possible sites for interaction were detected in the channel structure. MolDock Optimizer was the utilized algorithm and docking was evaluated 20 times for each ligand, considering 1000 possible binding conformations. The best pose was selected for each phenol, based on their docking energy score (MolDock Score). Further analysis for the residues interacting with each molecule was done (ligand map > ligand energy inspector). The amino acid contribution to the hVDAC1–phenol interaction was depicted by creating a backbone visualization in the workspace.

### 4.10. Statistical Analysis

Prism 6 (GraphPad Prism software, La Jolla, CA) was employed to perform the analysis. Data are presented as the mean ± SD. One-way ANOVA and Tukey’s multiple comparison test were employed, unless otherwise noted. *p*-values are represented as * (*p* ≤ 0.05), ** (*p* ≤ 0.01), *** (*p* < 0.001) and **** (*p* ≤ 0.0001).

## Figures and Tables

**Figure 1 ijms-22-00204-f001:**
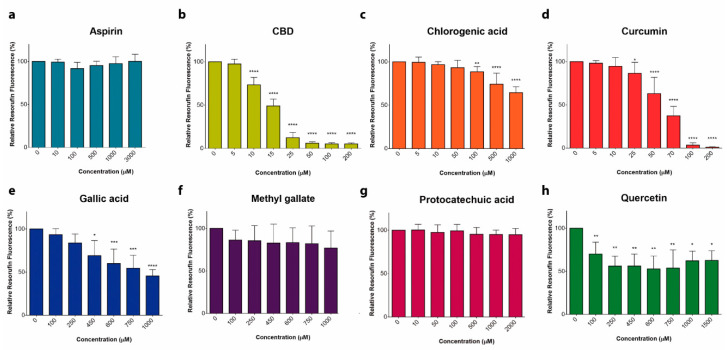
Cytotoxic effect of phenols on leukemic cells. (**a**–**f**) Cytotoxic effect of phenolic compounds was estimated by means of resazurin-based metabolic assay. T-ALL cells (Jurkat) were incubated in the presence or absence of aspirin (**a**), cannabidiol, CBD (**b**), chlorogenic acid (**c**), curcumin (**d**), gallic acid (**e**), methyl gallate (**f**), protocatechuic acid (**g**) and quercetin (**h**), for 24 h. Resorufin fluorescence was measured and normalized to untreated cells. Data are mean ± SD (*n* = 9 from three independent experiments; * *p* < 0.05; ** *p* < 0.01; *** *p* < 0.001; **** *p* < 0.0001; one-way ANOVA). Non-linear fit of the dose-dependence yields the following IC50 values (in µM): 12.1 for CBD and 36.5 for curcumin.

**Figure 2 ijms-22-00204-f002:**
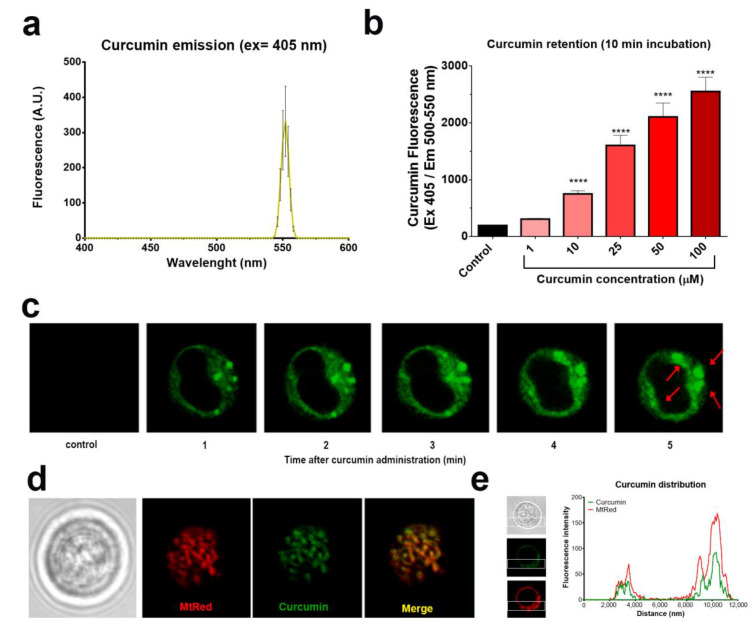
Curcumin is rapidly taken up by mitochondria in T-ALL cells. (**a**) Curcumin (50 µM) autofluorescence estimated by spectrofluorometry. (**b**) Dose-dependent uptake of curcumin by Jurkat cells. Data are mean ± SD (*n* = 12 from three independent experiments; **** *p* < 0.0001; one-way ANOVA). (**c**) Time course of curcumin (25 µM) uptake (green) by Jurkat cells. Red arrows indicate puncta of curcumin accumulation. (**d**) Co-localization of curcumin (25 µM, 10 min incubation) with the mitochondrial-selective fluorophore Mitotracker Red (MtRed). (**e**) Spatial distribution of curcumin and MtRed fluorescence intensity along the cell axis within a single Jurkat cell.

**Figure 3 ijms-22-00204-f003:**
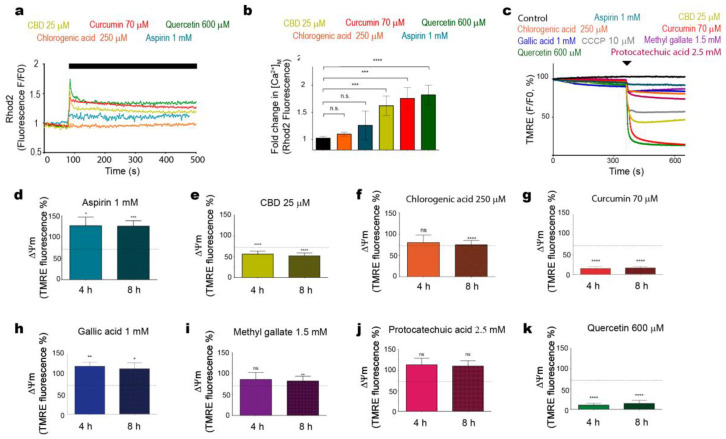
Effects of phenolic compounds on mitochondrial Ca^2+^ and ∆Ψm. (**a**) Jurkat cells stained with mitochondrial Ca^2+^ indicator Rhod-2 AM (2 µM) were treated with curcumin (red trace), CBD (yellow trace), aspirin (blue trace), chlorogenic acid (orange trace) or quercetin (green trace). The effect of phenols on Rhod-2 fluorescence was evaluated by spectrofluorometry and normalized to initial fluorescence of untreated cells. Traces represent the average of three samples from independent experiments ± SD. (**b**) The amplitude of [Ca^2+^]_m_ response (peak-[Ca^2+^]_m_ level prior drug administration) was averaged and plotted as mean ± SD for each phenol and compared to unstimulated cells (*n* = 6 from three independent experiments; ns: not significant; **** *p* < 0.0001; one-way ANOVA). The immediate effect of phenols on ∆Ψ_m_ in tetramethylrhodamine ethyl ester perchlorate (TMRE)-stained Jurkat cells was evaluated by spectrofluorometry (**c**). Traces represent the average of three samples from independent experiments. (**d**–**k**) The effect of aspirin (d), CBD (**e**), chlorogenic acid (**f**), curcumin (**g**), gallic acid (**h**), methyl gallate (**i**), protocatechuic acid (**j**) or quercetin (**k**) on ∆Ψm in TMRE-stained Jurkat cells at 4 and 8 h of treatment. Data are compared with a positive control (CCCP, 10 µM, 4 h, dashed line). TMRE fluorescence intensity in untreated cells is taken as 100%, data are mean ± SD (*n* = 9 from three independent experiments; ns: not significant; * *p* < 0.05; ** *p* < 0.01; *** *p* < 0.001; **** *p* < 0.0001; one-way ANOVA).

**Figure 4 ijms-22-00204-f004:**
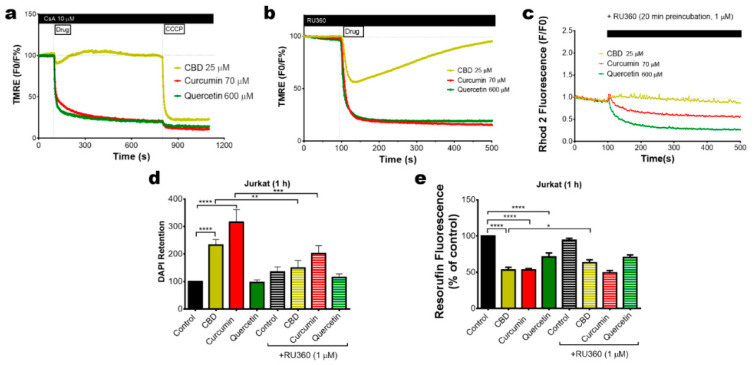
Pharmacology of the mitochondrial Ca^2+^ and ∆Ψ_m_ changes induced by cytotoxic phenols. (**a**) The effect of the mPTP inhibitor (CsA, 20 min preincubation) on ∆Ψ_m_ in TMRE-stained Jurkat cells, treated with phenols. Traces represent the average of three independent experiments. CCCP (10 µM) was added as a control at the end of each experiment. (**b**) Effect of Ru360 (1 µM, 20 min preincubation) on ∆Ψ_m_ in TMRE-stained Jurkat cells, treated with phenols. Traces represent the average of three independent experiments. (**c**) Effect of the MCU inhibitor Ru360 preincubation (1 µM) over the mitochondrial Ca^2+^ changes, induced by phenols, in Jurkat cells stained with Rhod-2 (2 µM). Traces represent the average of three independent experiments. Effect of Ru360 (1 µM, 20 min preincubation) on the cell death (**d**) or metabolism (**e**) of Jurkat cells, treated with CBD (25 µM), curcumin (70 µM) or quercetin (600 µM). Bars represent the average of three independent experiments ± SD, analyzed by a one-way ANOVA test (* *p* < 0.05; ** *p* < 0.01; *** *p* < 0.001; **** *p* < 0.0001).

**Figure 5 ijms-22-00204-f005:**
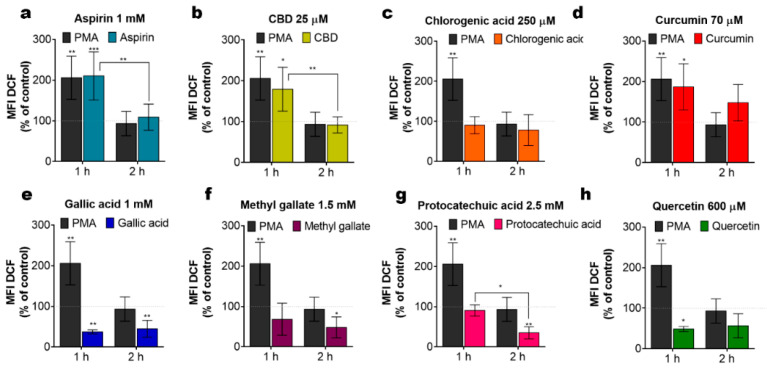
Intrinsic ROS levels in Jurkat cells, treated with different phenolic compounds. (**a**–**h**) Cells were stained with the indicator of ROS activity 2′, 7′-Dichlorodihydrofluorescein diacetate, DCF-DA (5 µM, 30 min) at 1 h or 2 h of treatment. DCF fluorescence was collected by flow cytometry (10,000 events per sample) and mean fluorescence intensity (MFI) was determined. The MFI value of the untreated cell population was taken as 100% (dotted line). Treatment with phorbol 12 myristate 13 acetate, PMA (5 µM) was used as a positive control. Data are mean ± SD (data from at least three independent experiments; * *p* < 0.05; ** *p* < 0.01; *** *p* < 0.001; two-way ANOVA).

## Data Availability

The data presented in this study are available in the article and supplementary materials.

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
