# Peer review of "Phenolic Compounds Cannabidiol, Curcumin and Quercetin Cause Mitochondrial Dysfunction and Suppress Acute Lymphoblastic Leukemia Cells"

_ijms, 2020, doi:10.3390/ijms22010204_

Round 1

Reviewer 1 Report

The manuscript by Olivas-Aguirre et al. entitled “Phenolic compounds cannabidiol, curcumin and quercetin cause mitochondrial dysfunction and suppress acute lymphoblastic leukemia cells” describes the cytotoxic effects of several phenolic compounds against human T-ALL cell line (Jurkat) and the mechanism insight into the anti-T-ALL activity with respect to ROS generation, electrical potential difference alterations of mitochondria, and Ca2+ homoeostasis. Several experiments have been carefully designed and carried out revealing, the correlation between the antileukemic effect of the most promising compounds (i.e. cannabidiol and curcumin) with the mitochondrial Ca2+ overload. The manuscript is convincing, well written and it offers new suggestions in an important scientific field. Taking into consideration all above mentioned, I would recommend the acceptance for publication of this manuscript in the International Journal of Molecular Sciences after the following Minor Revisions:

  • Line 51 “pK” should be pKa;
  • Line 61 specify the acronym VDAC;
  • Line 81 delete the round bracket;
  • Line 107 “Jurkat cells populations”;
  • Figure 3 increase font in all graphs;
  • Line 220 “..different region the in the pore wall..”.

Author Response

Reviewer 1

  • Line 51 “pK” should be pKa;
  • Line 61 specify the acronym VDAC;
  • Line 81 delete the round bracket;
  • Line 107 “Jurkat cellspopulations”;
  • Figure 3 increase font in all graphs;
  • Line 220 “..different region the in the pore wall..”.

All comments are gratefully acknowledged, and respective corrections were made.

Reviewer 2 Report

The authors investigated cytotoxic activity  of a series of phenolic compounds in T-ALL derived Jurkat cell line, suggesting that  natural phenols, as CBD and curcumin have promising antileukemic effect.

The paper is very interesting, experiments well conducted, methods clearly explained.

I have just few questions:

  1. They demonstrated that the anti-leukemic effect of curcumin and quercetin depended from the direct mitochondrial uncoupling and from mitochondrial Ca2+ overload. Have they any idea on the role of ABC proteins, which are also curcumin and quercetin substrates in determining the modification of intracellular ion channels?
  2. There was an effect/relationship with BCL2 function? If yes, can be hypothesized a synergistic effect with BLC2 inhibitors?
  3. Are the “active doses” compatible with “in vivo” use?

Author Response

Reviewer 2

  1. They demonstrated that the anti-leukemic effect of curcumin and quercetin depended from the direct mitochondrial uncoupling and from mitochondrial Ca2+ overload. Have they any idea on the role of ABC proteins, which are also curcumin and quercetin substrates in determining the modification of intracellular ion channels?

The reviewer has it right that the role of ABC proteins should be spelled. Indeed, curcumin and quercetin efficiently inhibit the multidrug resistance pumps, thus sensitizing cancer cells to chemotherapy. Respective note is introduced on the lines 235-238 and 2 new references were added.  

2. There was an effect/relationship with BCL2 function? If yes, can be hypothesized a synergistic effect with BLC2 inhibitors?

In our original Table S1 we have several entries, evidencing that indeed some of the tested phenolics change the Bcl-2 proteins balance, favoring the pro-apoptotic scenario.   

3. Are the “active doses” compatible with “in vivo” use?

This is a rather good point and it is worth to discuss. Indeed, there is a problem with curcumin and CBD bioavailability, so that their reported concentrations in serum are by one order of magnitude lower than the efficient ones in Fig. 1. We see two possible ways to overcome this problem. First is to develop the chemical derivatives of these compounds, which will act at lower concentration. For instance, drugs can be tagged in such a way that make them mitochondria targeted. Another non-exclusive approach is to get the advantage of the drugs synergism, which can substantially reduce their efficient antileukemic concentration. Currently we have obtained promising results with a combination of the CBD with some widely used anticancer drugs. At concentrations of the latter, which produced insignificant (<5%) cytotoxicity, the efficient antileukemic concentration of the CBD was decreased by several times. We have introduced the respective discussion on the lines 240-246 and added 3 new references.         

Minor issues in English were addressed.